# The transcriptomic response to heat stress of a jujube (*Ziziphus jujuba* Mill.) cultivar is featured with changed expression of long noncoding RNAs

Qing Hao[1]*, Lei Yang[1], Dingyu Fan[1], Bin Zeng[2,3], Juan Jin[1]

**1** Institute of Horticulture crops, Xinjiang Academy of Agricultural Sciences, Urumqi, China, **2** College of Forestry and Horticulture, Xinjiang Agricultural University, Urumqi, China, **3** Department of Crop Genetics and Breeding, Sub-branch of National Melon and Fruit Improvement Centre, Urumqi, China

* haoqingxj@sohu.com

**Data Availability Statement:** The RNA-seq data has been deposited in NCBI Gene Expression Omnibus (GEO) under accession code GSE136047.

## Abstract

Long non-coding RNA (lncRNA) of plant species undergoes dynamic regulation and acts in developmental and stress regulation. Presently, there is little information regarding the identification of lncRNAs in jujube (*Ziziphus jujuba* Mill.), and it is uncertain whether the lncRNAs could respond to heat stress (HS) or not. In our previous study, a cultivar (Hqing1-HR) of *Z. jujub*a were treated by HS (45°C) for 0, 1, 3, 5 and 7 days, and it was found that HS globally changed the gene expression by RNA sequencing (RNA-seq) experiments and informatics analyses. In the current study, 8260 lncRNAs were identified successfully from the previous RNA-seq data, and it indicated that lncRNAs expression was also altered globally, suggesting that the lncRNAs might play vital roles in response to HS. Furthermore, bioinformatics analyses of potential target mRNAs of lncRNAs with cis-acting mechanism were performed, and it showed that multiple differentially expressed (DE) mRNAs co-located with DElncRNAs were highly enriched in pathways associated with response to stress and regulation of metabolic process. Taken together, these findings not only provide a comprehensive identification of lncRNAs but also useful clues for molecular mechanism response to HS in jujube.

## Introduction

The long non-coding RNAs (lncRNAs), a class of transcripts from transcriptome without obvious open reading frames (ORFs), are defined as longer than 200 nucleotides at length [1], with a low conservation rate, low expression levels, and play important roles in multiple biological processes [2–4]. According to location and orientation to the nearest protein-coding transcripts, the lncRNAs are classified as intergenic lncRNAs, antisense lncRNAs, sense lncRNAs and intronic lncRNAs [5]. Generally, lncRNAs affect the biochemical and physiological processes by acting as molecular signals, guides, decoys or scaffolds in organisms [6], and their biological importance has led to great research interest in recent years [7]. Therefore, systematic identification and classification of lncRNAs has been conducted

**Funding:** This work was supported by the "National Natural Science Foundation of China" (Grant number: 31801815) and "Natural Science Foundation of Xinjiang China" (Grant number: 2019D01A64). The funders had no role in study design, data collection and analysis, decision to publish, or preparation of the manuscript.

**Competing interests:** The authors have declared that no competing interests exist.

in numerous species, with the development of transcriptome sequencing and computational methods [8].

Abiotic stresses, such as heat, drought, salinity, and cold, are major constraints to crop production and food security worldwide [9]. Currently, increasing evidence indicated that lncRNAs play vital roles in regulating responses to a variety of abiotic stressors [10]. In cassava (*Manihot esculenta*), 318 lncRNAs responsive to cold and/or drought stress were identified [11]. In cotton, it revealed that multiple lncRNAs possibly be associated with mediating plant hormone pathways in response to drought stress, and the lncRNA973 from upland cotton (*Gossypium hirsutum*) is involved with response to salt stress [12]. In pear (*Pyrus betulifolia*), a total of 14,478 lncRNAs were identified, and there were 251 drought-responsive ones [13]. In grapevine (*Vitis vinifera*), the expression dynamics of lncRNAs under cold stress was investigated using high-throughput sequencing, and hundreds of cold-related lncRNAs were found [14].

In particular, given the increasing evidence of climate change, the heat stress (HS) has become one of the most significant limiting factors to crop productivity, both qualitatively and quantitatively [15], and lncRNAs play vital roles in response to HS. In cabbage (*Brassica rapa*), under HS, 34 specifically expressed lncRNAs were identified, and 192 target genes were regulated by lncRNAs and most of them belonged to the heat-responsive genes [16]. In wheat (*Triticum aestivum*), 125 putative lncRNAs responsive to powdery mildew infection and heat were found [17].

The jujube (*Ziziphus jujuba* Mill.) (2n = 2x = 24), is one of the most important and popular fruit crops cultivated commercially in China [18]. The genome of the jujube has been sequenced recently [19, 20], but no systematic identification of lncRNAs and their responses to HS has been reported. Turpan is one of northeastern cities of in Xinjiang province of China, and has a unique temperate continental arid desert climate, with bright sunshine, high temperatures, and large day-night differences in temperature [21]. A putative heat-resistant jujube cultivar (Hqing1-HR) in an orchard of Turpan was found by chance, and bred successfully in our laboratory. In order to explore useful clues to explain the heat-resistance of Hqing1-HR cultivar, the seedlings were treated by HS (45˚C) for 0, 1, 3, 5 and 7 days, respectively, and the leaf samples (HR0, HR1, HR3, HR5 and HR7) were collected accordingly. The RNA sequencing (RNA-seq) experiments and informatics analyses were performed, and it indicated that expression levels of multiple genes associated with heat-resistance were greatly changed by the HS [22]. However, it also remained uncertain whether the lncRNAs could respond to HS or not. In the current study, using previous RNA-seq data, we tried to first identify lncRNAs and explore whether the lncRNAs could respond to HS or not. Here, we report the results.

## Materials and methods

### Data collection

The RNA-seq data has been deposited in NCBI Gene Expression Omnibus (GEO) under accession code GSE136047.

### LncRNA identification and classification

After discarding adapters and reads with low quality, the remained high quality clean reads were used for subsequent analyses. By using TopHat 2 software [23], these reads were mapped to the jujube genome. Only mapped reads with one genomic location were chose for further analysis. FPKM (fragments per million reads) was applied to measure the expression level of lncRNAs.

Novel transcripts were detected through Cufflinks and Cuffcompare [24], and the background noise was filtered according to coverage (>1), length (>200), FPKM (>0.5) and status threshold (OK) [24]. The coding potential capability of the novel transcripts was evaluated with the Coding Potential Calculator (value< 0) [25]. Class code 'u' represented long intergenic noncoding RNAs (lincRNAs). There is a threshold that lincRNAs must be more than 2000 bp away from the neighboring protein coding genes. The 'x' and 'i' represented long noncoding natural antisense transcripts (lncNATs) and intronic transcripts, respectively.

## Identification of common and specific lncRNAs

In order to compare the loci of lncRNAs from different sequencing samples, all separate transcriptome gtf files were merged into one file using Cuffmerge with the parameter–g [26]. The class code 'u' indicated specific lncRNAs. Moreover, these similar sequences were removed for ensuring the reliability of identified specific lncRNAs on basis of the reciprocal BLASTN results with an E threshold value (< 1e-10). The class code ' = ' represented core lncRNAs between *Z. jujuba*, which share completely equal loci. Reciprocal BLASTN analysis (E value < 1e-10) was also conducted to improve the confidence of core lncRNAs, and only those sequences with high similarity were kept for downstream analyses.

## Expression and functional analysis

To evaluate the pattern of lncRNA expression and explore differentially expressed (DE) lncRNAs or genes, the edge R package [27] which is specifically designed to analyze differential expression of genes using RNA-seq data, was used. The lncRNAs with FPKM < 0.1 in every sample were discarded before analysis. To explore DE lncRNAs or DE genes (DEGs), fold change (FC) ($\geq$2 or $\leq$0.5) and false discovery rate (FDR) ($\leq$0.01) cutoffs were used.

The putative functions of DE lncRNAs were explored by predicting their target genes of lncRNAs in cis manner. The genes which located at 100 kb upstream or downstream of lncRNAs were defined as target genes [28, 29].

To explore gene functions and measure the functional category distribution frequency, Gene Ontology (GO) analyses were carried out by using KOBAS bioinformatics resources [30]. Networks were constructed by calculating Pearson's correlation coefficients (PPCs) for genes and lncRNAs, and Cytoscape (v3.5.1) was used to display the co-expression network [31]. The distribution of lncRNAs in the *Z. jujuba* genome was plotted with circos software [32], and the venn diagrams were built using an online available tool (http://bioinfogp.cnb.csic.es/tools/venny/).

## Validation of RNA-seq by Real Time RT-PCR

To elucidate the validity of the RNA-seq data, quantitative real-time PCR (qPCR) was implemented for some DE lncRNAs. The information of primers is presented in Table 1. The same RNA samples for RNA-seq were used for qPCR. In each sample, l μg of RNA was reversely transcribed using the PrimeScript[TM] RT Reagent Kit (Takara, Dalian, China) according to the manufacturer's instructions. The qPCR was conducted on the Bio-Rad S1000 with Bestar SYBR Green RT-PCR Master Mix (DBI Bioscience, Shanghai, China). The PCR conditions are as follows: denaturing at 95˚C for 5 min, 35 cycles of denaturing at 94.5˚C for 25 s, annealing and extension at 60˚C for 1 min. The PCR amplifications were implemented in triplicate for each sample. Relative expression level of lncRNAs was calculated using the Livak and Schmittgen $2^{-\Delta\Delta Ct}$ method [33], normalized with the reference gene *Actin 3*.

**Table 1. The gene, lncRNA and primers used for qRT-PCR experiments.**

| LncRNA/Gene | Forward primer (5'-3') | Reverse primer (5'-3') |
|---|---|---|
| MSTRG.7381.6 | GGGCATCGCATTACCACATA | AACTTAGCAGGCAGCAGCAAC |
| MSTRG.20225.7 | GCGATCTCCAATCCAAGCAG | ATCGCCAGATGCAGTCCCA |
| MSTRG.36975.1 | CCAGTTGAGGTGCCCAATAAAGT | CCACCACTGTTTAGCCGTCAT |
| MSTRG.25280.9 | GGAGGGATGTTGTTGAGGGAG | GCTTGTTGGGTGCTTATTCTTG |
| ACT3 | GAAGCAACTGGCAACTAAGGC | CGAACAGACCGACCAAGTAAGC |

### Statistical analysis

The all data were analyzed statistically using Microsoft Excel (2010). All values were presented as mean ± standard deviation. For comparison between two groups, the significance of differences between means was determined by Student's t-test (paired). A value P<0.05 was considered to be statistically significant.

## Results

### Genome-wide identification and characterization of lncRNAs

We previously obtained a putative heat-resistant cultivar (Hqing1-HR) of *Z. jujuba*, and collected seedling samples on day 0, 1, 3, 5 and 7, post heat treatment at 45°C, respectively. Accordingly, 15 cDNA libraries (HR0-a, HR-0-b, HR0-c; HR1-a, HR1-b, HR1-c; HR3-a, HR3-b, HR3-c; HR5-a, HR5-b, HR-5-c; HR7-a, HR7-b, HR7-c) were prepared for RNA-seq, which composed three biological replicates at each time point. These RNA-seq data can be obtained from GEO under accession code GSE95203.

A total of 8260 predicted lncRNAs were identified successfully from the 15 transcriptome datasets by using an integrated approach (S1 Table). Among these putative lncRNAs, 51% (4632) were long intergenic noncoding RNAs (lincRNAs), 32.8% (2713) were natural antisense lncRNAs, 3.2% (263) were intronic lncRNAs, and 7.9% (630) were sense lncRNAs. The distribution of these lncRNAs in the *Z. jujuba* genome was plotted in a circus (Fig 1A). The global expression pattern of lncRNAs differed obviously at each time point (Fig 1B), demonstrating that HS greatly influenced the expression pattern of lncRNAs.

The mean length of lncRNA transcripts was shorter than that of mRNAs (931 bp for lncRNAs and 2078 bp for mRNAs) (Fig 1C), and the expression levels of lncRNAs were lower than expression levels of mRNAs (t-test p value = 0, Fig 1D). The lengths of lncRNAs ranged from 201–8199 bp, but more than 69% of lncRNAs were between 201 and 1000 bp (Fig 1E). About 63.6% of lncRNAs only had two exons, and 36.4% had more than two exons (Fig 1F). From the boxplots of the 15 samples, no difference was found in these 5groups (Fig 1G).

### Exploration of HS-induced DE lncRNAs

Using the edge R package [27], 193, 295, 458 and 250 DE lncRNAs (FDR $\leq$ 0.01, $|\log_2 FC| \geq 1$) were found in HR1 *VS* HR0, HR3 *VS* HR0, HR5 *VS* HR0 and HR7 *VS* HR0, respectively, with the highest or lowest FC being $2^{13.42}$ and $2^{-10.92}$ (Fig 2A) (S2 Table), indicating that the expression levels of lncRNAs were greatly influenced by HS (Fig 2B). In HR1 *VS* HR0, 118 up-regulated and 75 down-regulated lncRNAs were found. 222 up-regulated and 73 down-regulated lncRNAs were identified in HR3 *VS* HR0. There were 214 up-regulated and 244 down-regulated lncRNAs in HR5 *VS* HR0. In HR7 vs. HR0, 134 up-regulated and 116 down-regulated lncRNAs were discovered. It indicated that HS had effect on promoting and inhibiting the transcription of numerous lncRNAs.

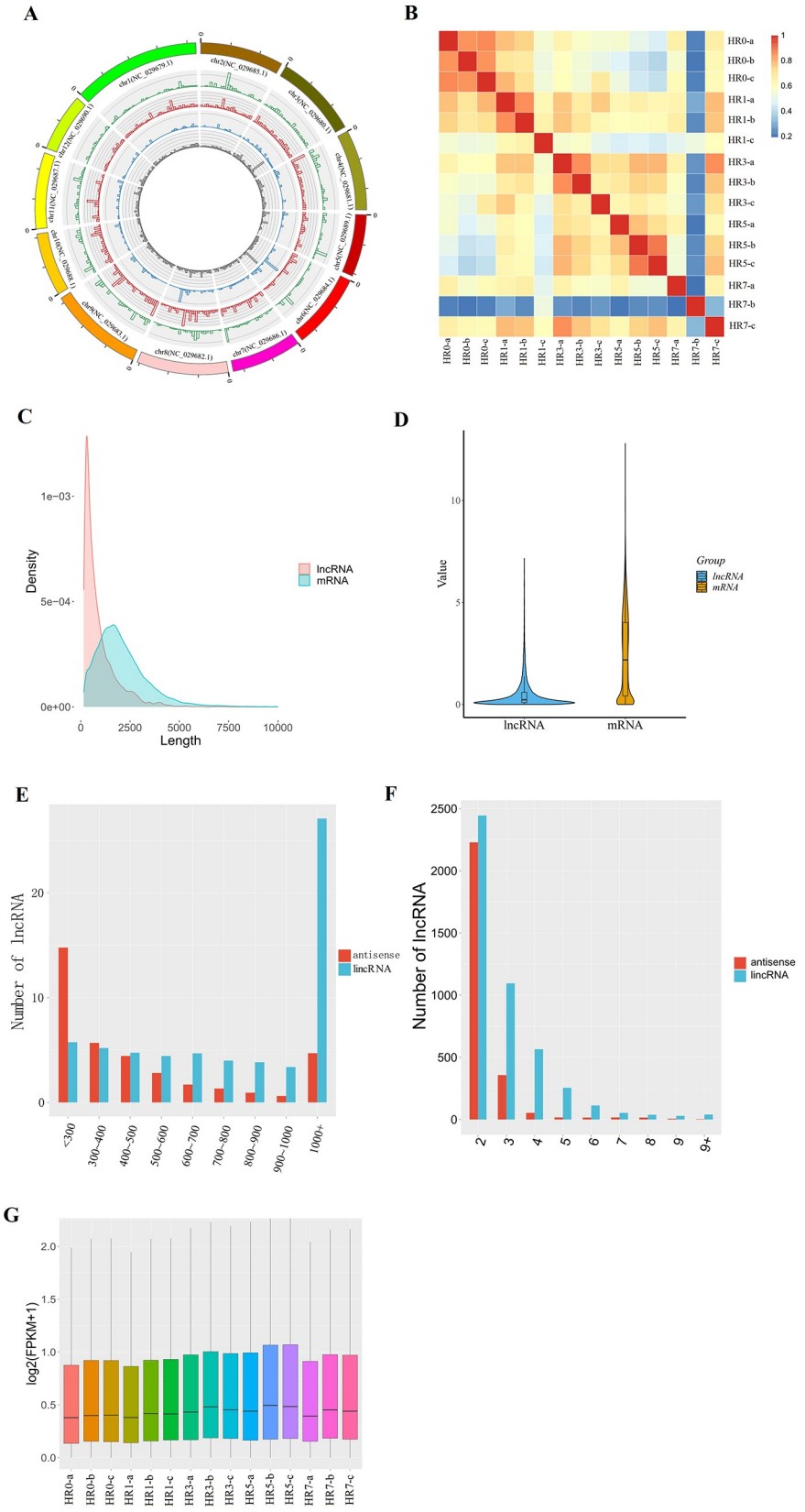

**Fig 1. Characterization of predicted lncRNAs in *Ziziphus jujuba*.** (A) Circos plot showing the genomic distribution of lncRNA clusters by distance. The green circle represents sense lncRNAs, while the red circle indicates long intergenic noncoding RNAs. The blue circle indicates intronic lncRNAs and the grey circle represents antisense lncRNAs. (B) Hierarchical clustering heatmap showing the distinct lncRNA expression patterns of heat-treated and control samples. (C) Length distribution of all lncRNA and mRNA transcripts. (D) Violin plot showing lower expression levels of lncRNAs compared with mRNAs. (E) Barplot showing the length distributions of lincRNAs and antisense lncRNAs. (F) Exon distribution of lincRNA and antisense lncRNAs. (G) Boxplot showing the FPKM (fragments per kilobase per million mapped fragments) distribution of lncRNAs from the 15 samples of jujube.

Moreover, there were only 40 common DE lncRNAs were found in four comparisons (S3 Table), demonstrating that lncRNA expression in response to HS was different in different time points (Fig 2C).

The genome of jujube has 12 chromosomes, and the distribution of DE lncRNAs on chromosomes was investigated. In the four comparisons, HS-induced DE lncRNAs were distributed across every chromosome (Fig 2D). In the HR1 *VS* HR0, more DE lncRNAs were found on chromosomes 02 and 06; while more DE lncRNAs were found on chromosomes 02, 04, 08 and 09 in HR3 *VS* HR0. In the HR5 *VS* HR0, more DE lncRNAs were found on chromosomes 02, 04, 05, 06 and 08; and more DE lncRNAs were found on chromosomes 02 and 06 in HR7 *VS* HR0, suggesting the non-uniform distribution of DE lncRNAs.

## DE lncRNAs regulate the response to HS

LncRNAs may function to modulate the transcription of nearby genes in a cis-acting manner [34], and cis-acting regulation between lncRNAs and mRNAs also occurs. In order to explore the function of these DE lncRNAs, the search for cis-acting target mRNAs was performed with a threshold distance of 100 kb between mRNAs and lncRNAs, and 1911, 3474, 4066 and 1996 putative target mRNAs were identified for the DE lncRNA from HR1 *VS* HR0, HR3 *VS* HR0, HR5 *VS* HR0 and HR7 *VS* HR0 (S4 Table), respectively.

Subsequently, we explored 2266, 4907, 6120 and 2894 DEGs in HR1 *VS* HR0, HR3 *VS* HR0, HR5 *VS* HR0 and HR7 *VS* HR0, respectively (S5 Table). Moreover, it was found that 150, 596, 877 and 200 cis-acting target mRNAs were overlapped with DEGs in HR1 *VS* HR0, HR3 *VS* HR0, HR5 *VS* HR0 and HR7 *VS* HR0 (Fig 3A, 3C, 3E and 3G; S6 Table), respectively, suggesting that multiple DE lncRNAs might play important roles in regulating the gene expression under HS.

To identify the pathways in which the DEGs putatively regulated by DE lncRNAs in a cis-acting manner were mainly involved, GO enrichment analysis was conducted. It showed that 38, 73, 73 and 35 biological process terms were identified in HR1 *VS* HR0, HR3 *VS* HR0, HR5 *VS* HR0 and HR7 *VS* HR0 (P<0.05) (S7–S10 Tables), respectively.

In HR1 *VS* HR0, "response to biotic stimulus (GO: 0009607)", "defense response (GO: 0006952)", "cellular response to heat" (GO: 0034605), and "response to extracellular stimulus (GO: 0009991)" were found (Fig 3B and S7 Table); while "response to heat (GO: 0009408)" and "response to red light (GO: 0010114)" could be identified in HR3 *VS* HR0 (Fig 3D and S8 Table). The "cellular response to water deprivation (GO: 0042631)","response to red light (GO: 0010114)", "cold acclimation (GO: 0009631)" and "systemic acquired resistance, salicylic acid mediated signaling pathway (GO: 0009862)" were found in HR5 *VS* HR0 (Fig 3E and S9 Table); and "response to stress (GO: 0006950)", "response to extracellular stimulus (GO: 0009991)","response to heat (GO: 0009408)", "leaf senescence (GO: 0010150)" and "systemic acquired resistance, salicylic acid mediated signaling pathway (GO: 0009862)" were identified in HR7 *VS* HR0 (Fig 3G and S10 Table). It suggested that the lncRNAs might respond to HS by modulating the expression levels of genes associated with response to HS and contribute to the heat-resistance of the Hqing1-HR cultivar.

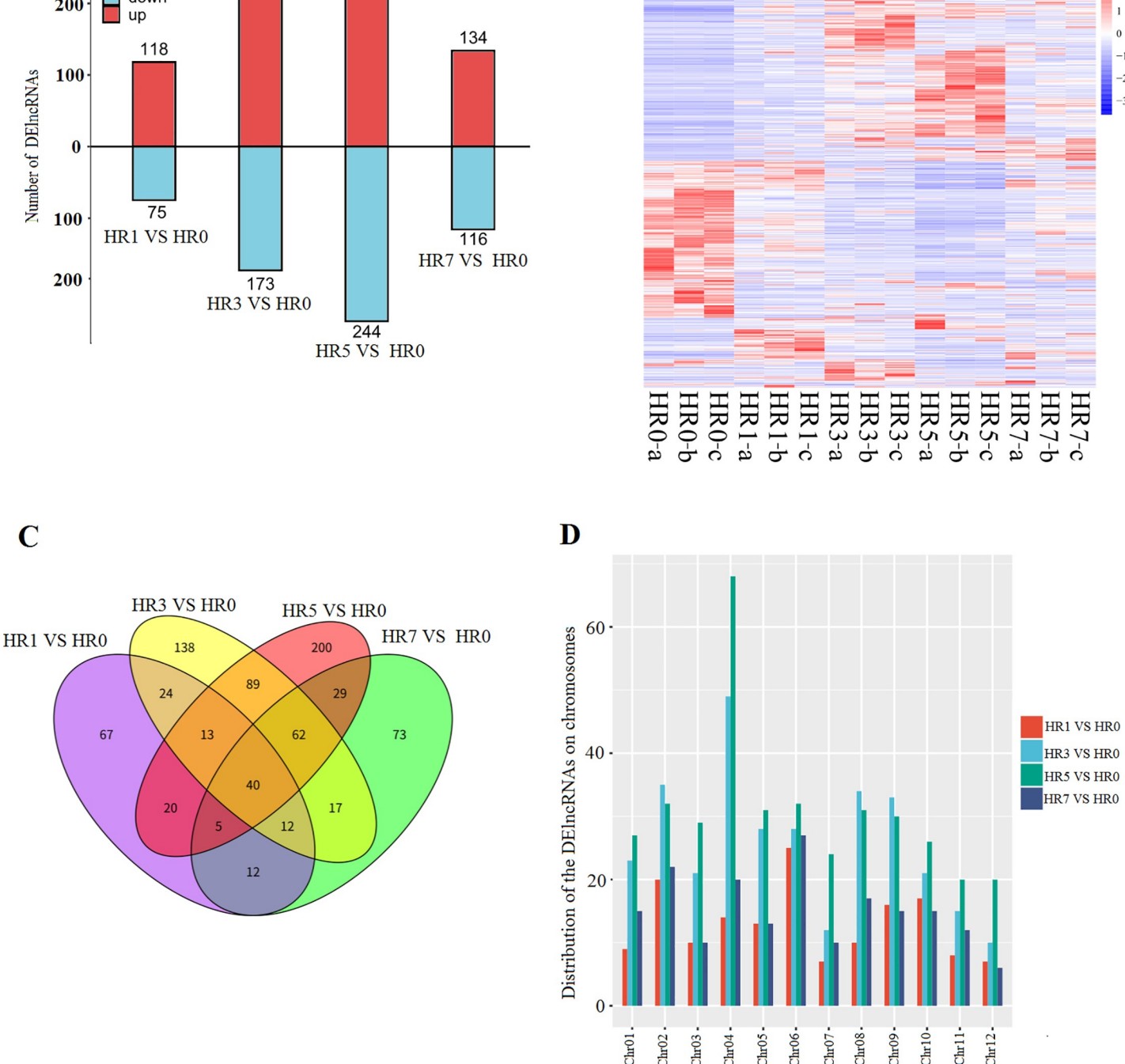

**Fig 2. Exploration of differentially expressed (DE) lncRNAs in response to heat stress (HS).** (A) The number of up-regulated and down-regulated DE lncRNAs from the four comparisons (HR1 *VS* HR0, HR3 *VS* HR0, HR5 *VS* HR0 and HR7 *VS* HR0). (B) Hierarchical clustering heatmap of all DE lncRNAs from the four comparisons. (C)Venn diagram showing the overlapping DE lncRNAs from the four comparisons. (D) Barplot showing the distribution of DE lncRNAs on chromosomes from Hqing1-HR cultivar of *Z. jujuba*.

**A** 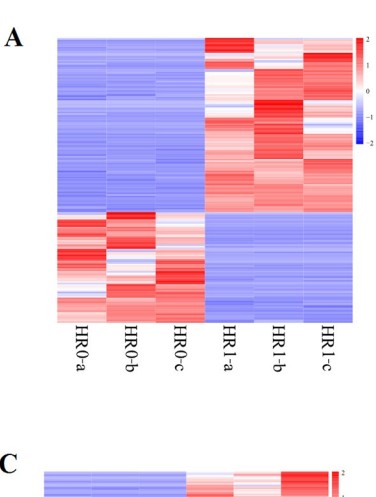

**B** 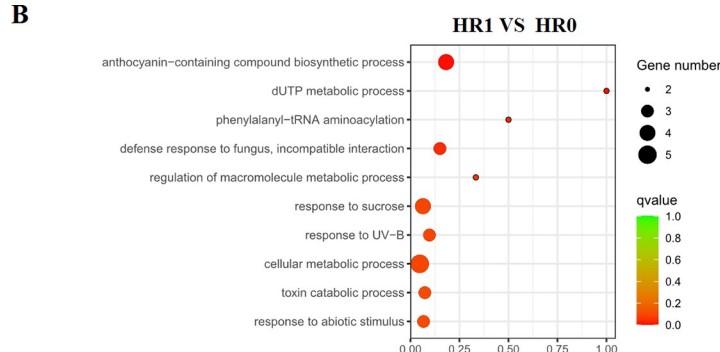

**C** 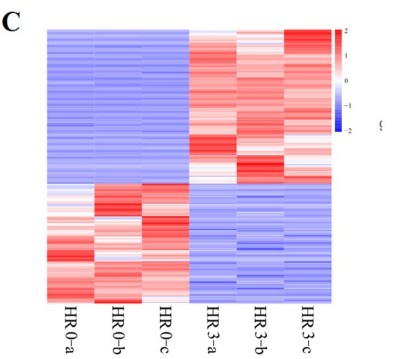

**D** 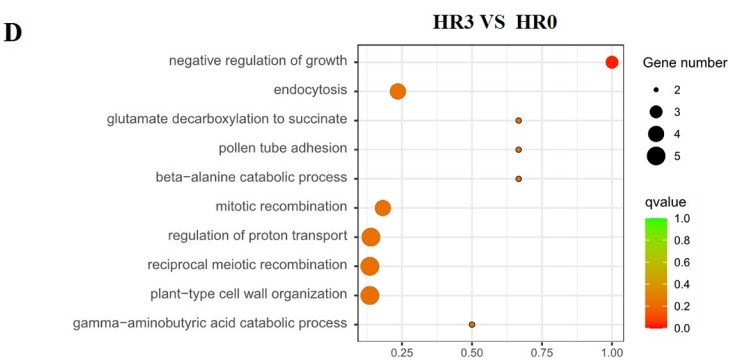

**E** 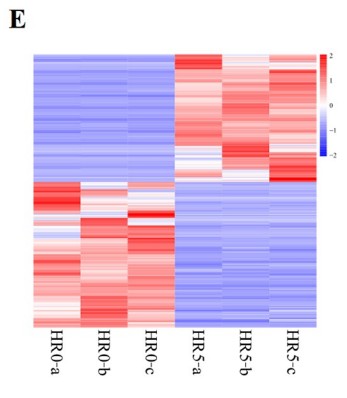

**F** 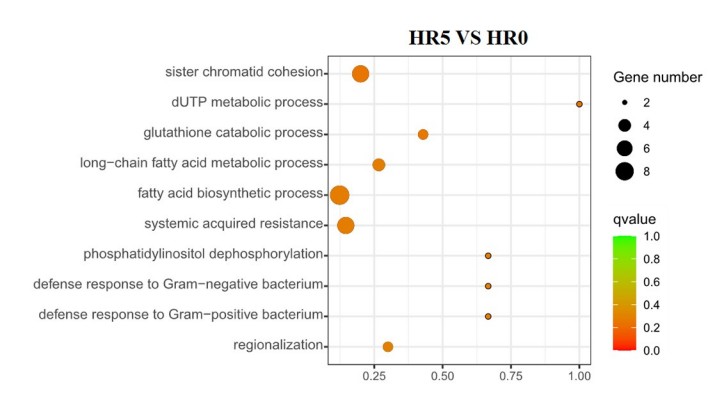

**G** 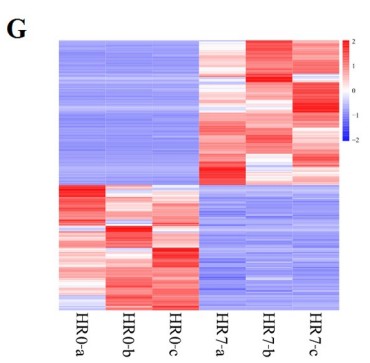

**H** 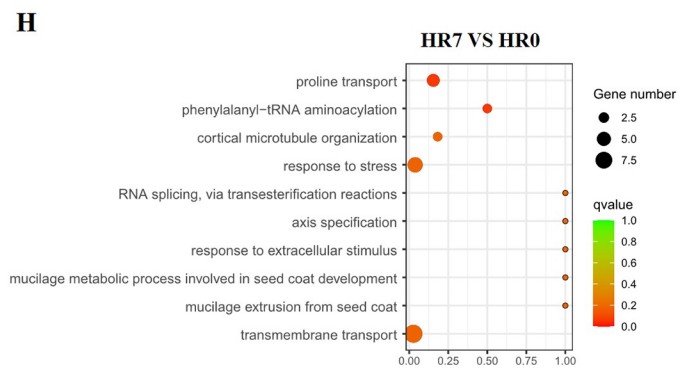

**Fig 3. Search for the DE mRNAs nearby the DE lncRNA and functional analysis.** (A), (C), (E) & (G) The hierarchical clustering heatmaps of DE mRNAs nearby the DE lncRNA from HR1 *VS* HR0, HR3 *VS* HR0, HR5 *VS* HR0 and HR7 *VS* HR0, respectively. (B), (D), (F) & (H) The functional analyses of the mRNAs nearby the DE lncRNAs from four comparisons above. Only the top 10 terms are listed here.

Moreover, multiple terms associated with metabolic process were also found in the four comparisons above, suggesting the multiple lncRNAs could respond to HS by regulating the expression levels of genes involved with metabolism (Fig 3 and S7–S10 Tables).

## RT-qPCR validation for RNA-seq

To validate the reliability of the transcriptome gene expression profiles, 4 DE lncRNAs were randomly selected for expression analysis through RT-qPCR (Fig 4). The expression patterns shown in the RT-qPCR results (Fig 4B) were consistent with RNA-seq results (Fig 4A). Moreover, the results of RT-qPCR experiments were consistent with the results from RNA-seq (Fig 4A) with PCCs higher than 0.9 (Fig 4C), suggesting the accuracy of these data.

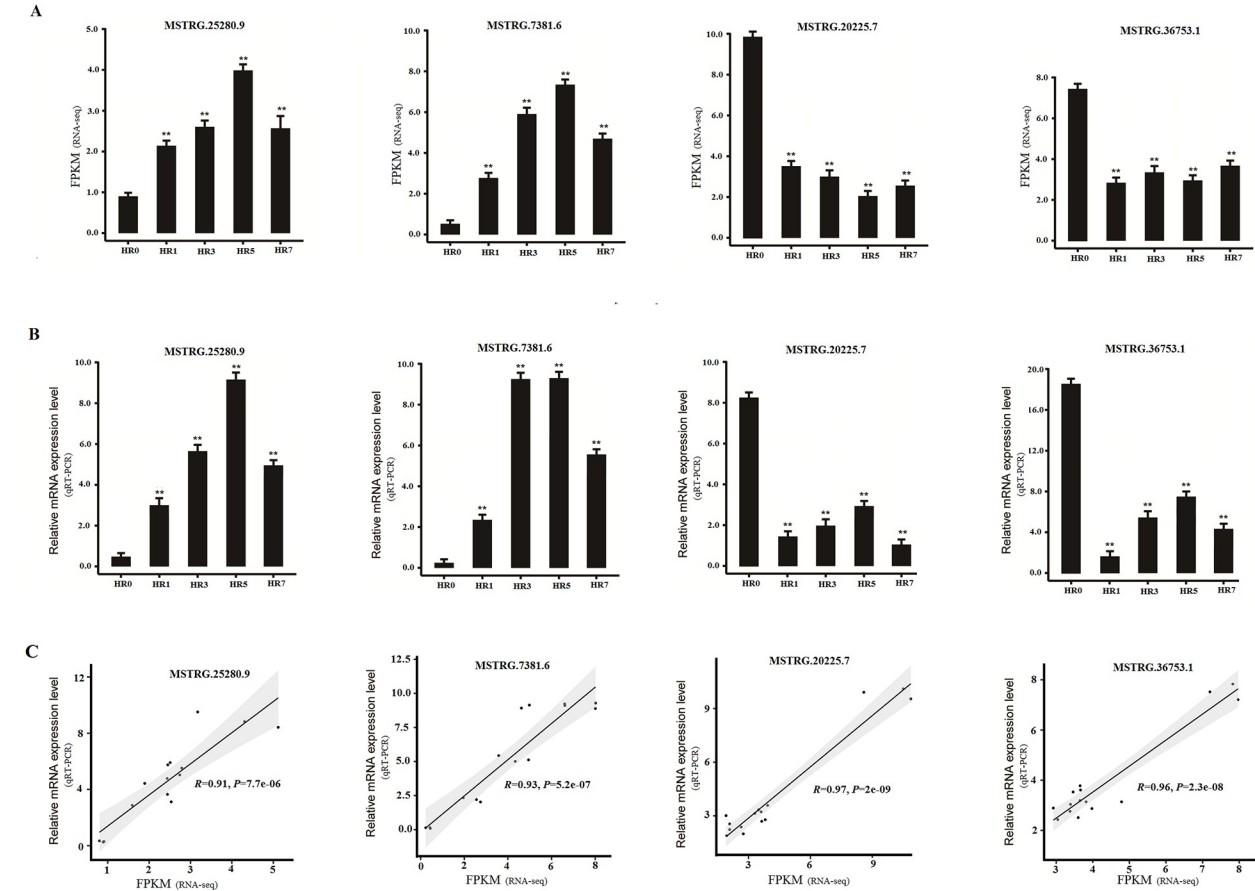

**Fig 4. Quantitative real-time PCR (qRT-PCR) validation of some differentially expressed (DE) lncRNAs obtained from RNA sequencing (RNA-seq).** (A) The relative expression levels of some DE lncRNAs were determined by RNA-seq. FPKM (fragments per kilobase per million mapped fragments) was used to calculate the expression levels of genes or lncRNAs. (B) The relative expression levels of the DE lncRNAs above were validated by qRT-PCR, normalized with the *ACT3* gene. Three replicates were used for RT-qPCR experiments. (C) The correlation analyses for the data between the qRT-PCR and RNA-seq using the Pearson's correlation coefficients (PCCs). Data represent the mean values ±SD. *, $P<0.05$; **, $P<0.01$, calculated with student's *t* test.

## Discussion

The jujube (*Z. jujuba*), one member of the Rhamnaceae family, is a popular fruit tree species with huge economic value [18, 35], and its domestication dated back to 7,000 years ago in China [20, 35]. Generally, the fruit of jujube is widely consumed as a food or food additive due to the high nutritional value [36], and has been introduced into more than 50 countries throughout the five continents [37]. It was found that more than 700 subspecies, varieties, and cultivars of jujube have been discovered or developed during the process of cultivation [38], providing the valuable resource of agriculture.

With the rapid and drastic changes in the climate, the food security in the world is being menaced [39], and it demonstrated that HS repeatedly reduced the yields of major crops such as wheat, rice (*Oryza Satiua*), maize (*Zea mays*) and soybean (*Solanum tuberosum*) [40], in addition to horticultural crops such as grapevine, almonds (*Amygdalus communis*), apples (*Malus pumila*), oranges (*Citrus reticulata*) and avocados (*Persea americana*) [41]. We also found that HS had bad effects on quantity and quality of jujube in Xinjiang province of China. So, searching and breeding heat-resistant cultivars might be one of feasible and important strategies to protect production and quality of crops, including jujube. For examples, the heat-resistant cultivars were found in some major crops, including rice (*O. sativa*) [42], maize [43], and wheat [44], but the heat-resistant cultivars of horticultural crops were seldom reported. In the summer of 2017, we found a putative heat-resistant cultivar (Hqing1-HR) of jujube in the Turpan city of China. To our knowledge, the Hqing1-HR might be the first heat-resistant jujube cultivar, which might be potentially beneficial for developing more heat-resistant cultivars in the future.

As sessile organisms, crops always induce serious molecular and physiological changes in response to HS [45], including rapid transcriptomic and metabolic adjustments [9, 46]. To explore the molecular mechanism of the heat-resistant jujube cultivar response to HS, the RNA-seq experiments were carried out [22], and it revealed that multiple DEGs were associated with HS. In fact, the noncoding RNAs (ncRNAs), including microRNAs (miRNAs), lncRNAs and circular RNAs (circRNAs), is also involve with HS in crops [47]. In the jujube, there were no studies, in which the systematic identification and classification of ncRNAs. In the current study, we first identified 8260 lncRNAs using the previous RNA-seq data, and the genomic distribution of lncRNA clusters on 12 chromosomes did not show any bias (Fig 1A). However, the distribution HS-induced DE lncRNAs on these chromosomes was non-uniform (Fig 2D), more DE lncRNAs were found on chromosome 02, 04, 06 and 08, suggesting there might be some genomic hotspots response to HS by regulating expression of lncRNAs. Albeit multiple DE lncRNAs were found, but only 40 common DE lncRNAs were identified (Fig 2C and S3 Table). Moreover, there were 67, 138, 200 and 73 unique DE lncRNAs were found in HR1 *VS* HR0, HR3 *VS* HR0, HR5 *VS* HR0 and HR7 *VS* HR0, respectively, suggesting that numerous lncRNAs might specifically respond to HS in every time points.

Furthermore, multiple putative cis-target DEGs of the DE lncRNAs were enriched in the pathways associated with response to HS and regulation of metabolic process, suggesting that some lncRNAs might contribute to the heat-resistance of this jujube cultivar (Fig 3 and S7–S10 Tables). In addition, to deeply explore the transcriptomic molecular mechanism of Hqing1-HR cultivar response to HS, the systematic identification and research for miRNAs and circRNAs should be implemented in the future.

## Supporting information

**S1 Table. The list of identified lncRNAs.**
(XLS)

**S2 Table. The list of the DE lncRNAs.**
(XLSX)

**S3 Table. The common DElncRNAs among the four comparisons.**
(XLS)

**S4 Table. The putative cis-target genes of the DE lncRNAs.**
(XLSX)

**S5 Table. The list of the DEGs.**
(XLSX)

**S6 Table. The list of DElncRNAs and corresponding putative cis-target DEGs.**
(XLSX)

**S7 Table. The functional analyses of cis-target DEGs of DE lncRNAs from HR1 VS HR0.**
(XLS)

**S8 Table. The functional analyses of cis-target DEGs of DE lncRNAs from HR3 VS HR0.**
(XLS)

**S9 Table. The functional analyses of cis-target DEGs of DE lncRNAs from HR5 VS HR0.**
(XLS)

**S10 Table. The functional analyses of cis-target DEGs of DE lncRNAs from HR7 VS HR0.**
(XLS)

## Acknowledgments

We thank Dr. Leilei Zhan for analyzing sequence data, and his helpful discussions.

## Author Contributions

**Conceptualization:** Lei Yang.

**Data curation:** Qing Hao, Lei Yang, Dingyu Fan, Juan Jin.

**Formal analysis:** Dingyu Fan, Bin Zeng, Juan Jin.

**Investigation:** Qing Hao.

**Project administration:** Qing Hao, Lei Yang.

**Validation:** Lei Yang, Dingyu Fan.

**Writing – original draft:** Qing Hao, Bin Zeng.

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
