## [Decision Letter · Decision Letter 0]

1 Jan 2021

PONE-D-20-29127

The transcriptomic response to heat stress of a jujube (Ziziphus jujuba Mill.) cultivar is featured with changed expression of long noncoding RNAs

PLOS ONE

Dear Dr. Hao,

Thank you for submitting your manuscript to PLOS ONE. After careful consideration, we feel that it has merit but does not fully meet PLOS ONE’s publication criteria as it currently stands. Therefore, we invite you to submit a revised version of the manuscript that addresses the points raised during the review process. Expression of language is very poor and needs improvement as pointed out by the reviewers.

We look forward to receiving your revised manuscript.

Kind regards,

Pradeep Sharma

Academic Editor

PLOS ONE

"This work was supported by the “National Natural Science Foundation of China”

(Grant number: 31801815) and “Natural Science Foundation of Xinjiang China”

(Grant number: 2019D01A64)."

6. Thank you for submitting the above manuscript to PLOS ONE. During our internal evaluation of the manuscript, we found significant text overlap between your submission and the following previously published works, some of which you are an author.

- https://www.nature.com/articles/s41598-017-17179-3

- https://www.frontiersin.org/articles/10.3389/fpls.2016.01213/full

- https://journals.plos.org/plosone/article?id=10.1371/journal.pone.0186681

- http://www.ablife.cc/wp-content/uploads/2017/11/Transcriptome-Analysis-of-Sheep-Oral-Mucosa-Response-to-Orf-Virus-Infection.pdf

Please revise the manuscript to rephrase the duplicated text, cite your sources, and provide details as to how the current manuscript advances on previous work. Please note that further consideration is dependent on the submission of a manuscript that addresses these concerns about the overlap in text with published work.

**Comments to the Author**

Reviewer #1: Hao et al, perform RNA -seq profiling of the jujibe plant following heat stress. They identify differentially expressed lncRNAs over a time course of heat stress. They investigate the possibility that some lncRNAs are correlated with their nehioring mRNAs and could be involved in cis regulation. They confirm the induction of a small number of lncRNAs identified in the RNA-seq by qPCR. This study provides a resource on genes dysregulated during heat stress in jujube. I have some comments below that if addressed will make this publication acceptable for publication in Plos one.

Major:

• The authors say 51% (4632) were long intergenic noncoding RNAs (lincRNAs), how are they defining intergenic. Is there a threshold for how far away the gene must be from a neighboring protein coding gene? (to ensure it is not simply an unnotated start site).

• When the authors say “In our previous study, we also explored the 2266, 4907, 6120 and 2894 DEGs in HR1 VS HR0, HR3 VS HR0, HR5 VS HR0 and HR7 VS HR0, respectively (Data not shown).” Is this a published study? If so please add the reference instead of data not shown. Overall this entire section as written is very confusing to me. Are the authors looking at the cis acting effects of the differentially expressed lncRNAs from table 1 or from another study entirely? I think it is from this study but as it is written right now   it is very confusing.

• The entire section on “DE lncRNAs regulate the metabolic process”. I am not sure how informative this section is to simply list out all of the metabolic processes. I don’t think outlining each one in necessary. This whole section could be summarized into a couple of sentences.

• Figure 2C. it would be useful to the reader to add the common 40 lncRNAs that are differentially expressed into a separate table.

• The authors say “To our surprise, it was found that 1911, 3474, 4066and 1996 cis-acting target mRNAs were overlapped with DEGs in HR1 VS HR0, HR3VS HR0, HR5 VS HR0 and HR7 VS HR0 (Fig 3 A, C, E and G)”. What do they mean as targets? I think it would be very useful to break it down into the mRNAs that show the same pattern of induction as the neighboring lncRNA and which are anticorrelated in expression level (where the mRNA is up and the lncRNA is down or vice versa). Placing this lists of potential cis regulators together within a table would be a good resource to accompany Figure 3.

• The authors say “Moreover, the results of qRT-PCR experiments were consistent with the results from RNA-seq (Fig. 4A) with Pearson correlation coefficients (PCCs) higher than 0.9, suggesting the accuracy of these data”. This data comparison is now shown. It would be useful to show the data and what they are measuring the PCCs within this figure.

Minor:

Define DE first before abbreviating

Esthetically it is nice to have font similar sizes between graphs within the same figure

Reviewer #2: The manuscript which is named “The transcriptomic response to heat stress of a jujube (Ziziphus jujuba Mill.) cultivar is featured with changed expression of long noncoding RNAs” analyzed the lncRNAs which might play an important role in response to HS in Chinese jujube. However, some modifications need to be done to improve the manuscript.

The most important question for this manuscript is the language. I strongly suggest the language of this manuscript should be polished by native speaker. For example, in the abstract, what is “not also, but also”. Introduction, “biotic stressors”、 “is involved response to”“was made to be”and so on. Discussion “Our found a putative cultivar (Hqing1-HR) and bred it in our laboratory” and so on, these sentences are difficult to understand.

Then the character of the cultivar Hqing1-HR should be described in the introduction part, why chose this material to do heat stress?

The results parts are superficial, after reading I do not get any interesting information. Also the discussion part should be rewrote and discuss more lncRNAs in Chinese jujube. The authors should explain the results and give the reasons.

---

## [Author Response · Author response to Decision Letter 0]

14 Mar 2021

Dear editors, 

Thank you for arranging a timely review for our manuscript (Manuscript ID: PONE-D-20-29127). We have carefully evaluated your critical comments and thoughtful suggestions, and improved the manuscript accordingly. We are glad to resubmit the MS right now.

The relevant comments and our responses are presented in a rebutal letter.

We look forward to hearing from you soon, and will further improve the MS according to your kind suggestions.

Best regards!

Qing Hao

---

## [Editor Report · Decision Letter 1]

23 Mar 2021

The transcriptomic response to heat stress of a jujube (Ziziphus jujuba Mill.) cultivar is featured with changed expression of long noncoding RNAs

PONE-D-20-29127R1

Dear Dr. Liu,

We’re pleased to inform you that your manuscript has been judged scientifically suitable for publication and will be formally accepted for publication once it meets all outstanding technical requirements.

Kind regards,

Pradeep Sharma

Academic Editor

PLOS ONE

---

## [Editor Report · Acceptance letter]

26 Mar 2021

PONE-D-20-29127R1 

The transcriptomic response to heat stress of a jujube (Ziziphus jujuba Mill.) cultivar is featured with changed expression of long noncoding RNAs 

Dear Dr. Hao:

I'm pleased to inform you that your manuscript has been deemed suitable for publication in PLOS ONE. Congratulations! Your manuscript is now with our production department. 

Kind regards, 

on behalf of

Dr. Pradeep Sharma 

Academic Editor

PLOS ONE